Research article

# Enzymes of physiological amyloidogenesis control pathological amyloid toxicity

Michael Bokros[1,2], Alex Grunfeld[1,2], Nathan C Balukoff[1,2], Jessica Bouviere[1,2], Eléonore Beurel[1,3], Stephen Lee[1,2]

**Physiological amyloidogenesis drives the formation of functional amyloids involved in various biochemical pathways. We recently showed that the RNA tailing and decay machinery controls the maturation of intracellular amyloid-like aggregates. This raises the question of whether enzymes that participate in the maturation of physiological amyloids are involved in pathological amyloidogenesis implicated in human proteopathies. Using *Caenorhabditis elegans* and mouse models of pathological amyloids, we show that manipulating the RNA tailing–decay axis alters the toxicity of *β*-amyloid and *α*-synuclein involved in Alzheimer's and Parkinson's diseases, respectively. The RNA tailing enzymes TENT4b and TENT2 protect against *β*-amyloid– and *α*-synuclein–induced toxicity by facilitating the formation of nontoxic amyloidogenic assemblies. In contrast, the RNA exonuclease Exosc10 potentiates pathological amyloid toxicity. Remarkably, Exosc10 depletion prevents cognitive decline and restores memory in two different mouse models of *β*-amyloid neurotoxicity. Taken together, these results suggest that pathways of physiological amyloidogenesis participate in pathological amyloid etiology.**

## Introduction

The discovery of proteins adopting the amyloid conformations in pathological conditions has tightly linked the amyloid fold to disease etiology. Indeed, more than 50 proteins have been implicated in proteopathic disorders because of their capacity to form amyloid aggregates (1). Meanwhile, accumulating evidence reveals that amyloid structures are not solely pathological; a growing number of proteins adopt this conformation to serve essential physiological roles (2, 3, 4, 5). The term functional amyloid was first introduced in 2000 by Wösten and de Vocht, who described fungal coat proteins (hydrophobins) forming classical cross-*β*-sheet amyloid folds to fulfill specific biological functions (6). Since then, functional amyloidogenesis has been documented across diverse biological systems, encompassing a wide range of physiological contexts. The amyloid conformation—defined by its highly ordered cross-*β*-sheet structure—confers exceptional stability, making it particularly advantageous in settings requiring molecular longevity. This includes roles in long-term memory storage (7, 8, 9, 10), protein storage reservoirs (11, 12), or cellular dormancy (13, 14, 15, 16, 17). Furthermore, the rapid and templated conversion of monomers into amyloid assemblies enables efficient signal propagation and amplification (18, 19, 20). These unique biophysical properties of amyloids are increasingly being harnessed for biomedical and biotechnological applications (21, 22) and have been hypothesized to be an origin of replicative life (23, 24, 25). Together, these insights suggest that amyloidogenicity, rather than being intrinsically deleterious, represents a conserved and versatile structural strategy across biology.

We recently identified a physiological amyloid–regulatory axis that relies on the RNA tailing TENT4b and the riboexonuclease RNA exosome (26). TENT4b synthesizes polyanionic RNA molecules that facilitate the formation of mature amyloidogenic aggregates. RNA exosome antagonizes this process by inhibiting intracellular amyloidogenesis, both within the nucleolus and in the cytoplasm. Given the role of TENT4b–RNA exosome axis in physiological amyloidogenesis, it raises the question whether these enzymes influence pathological amyloidogenesis involved in several human proteopathies. Here, we show that manipulating the RNA tailing/decay axis modulates amyloidogenic toxicity in multiple *Caenorhabditis elegans* and mouse models of pathological amyloids. Taken together, these results link mechanisms of physiological amyloidogenesis to pathological amyloid toxicity opening new avenues of investigation in the field of proteopathies.

## Results and Discussion

### TENT enzymes protect against *β*-amyloid and *α*-synuclein toxicity

Previous work suggests opposing roles of TENT4b and RNA exosome in functional amyloid maturation (26). TENT4b facilitates

[1]Department of Biochemistry and Molecular Biology, Miller School of Medicine, University of Miami, Miami, FL, USA    [2]Sylvester Comprehensive Cancer Center, Miller School of Medicine, University of Miami, Miami, FL, USA    [3]Department of Psychiatry and Behavioral Sciences, Miller School of Medicine, University of Miami, Miami, FL, USA

Correspondence: stephenlee@med.miami.edu

amyloid formation, whereas the RNA exosome suppresses intracellular amyloidogenesis. Depletion of the RNA exosome catalytic subunit EXOSC10 removes this inhibition, enabling the formation of mature amyloidogenic aggregates in different subcellular compartments. The question is whether altering the activity of these physiological amyloidogenesis regulatory enzymes has an impact on pathological amyloidogenesis involved in human proteopathies.

To address this question, we first chose the classical *C. elegans* strain (CL2006) that constitutively expresses Alzheimer's disease (AD)–relevant human $\beta$-amyloid[1-42] [27]. The CL2006 transgenic *C. elegans* undergo age-dependent paralysis owing to the accumulation of $\beta$-amyloid[1-42] in body wall muscle cells [28] and have been used extensively to uncover biochemical pathways and small molecules involved in $\beta$-amyloid[1-42] intracellular toxicity [29]. This model was chosen because the toxicity timeline allows to accurately measure both the accelerated and decreased toxicity resulting in the manipulated TENT/RNA exosome system. In addition, other systems that induce $\beta$-amyloid[1-42] expression by heat-induced stabilization of the transgene introduce confounding effects of the RNA tailing and degradation machinery on $\beta$-amyloid[1-42] mRNA. *TENT4a* and *TENT4b* are conserved in *C. elegans* as the *GLD-4* homolog [30, 31, 32]. RNAi-mediated depletion of TENT4[GLD-4] significantly accelerated paralysis (Figs 1A and S1A) associated with the accumulation of $\beta$-amyloid species believed to be toxic [28] (Fig 1B), without affecting $\beta$-amyloid[1-42] transgene mRNA levels (Fig S1B). Paralysis was not observed in nontransgenic worms (N2) depleted of TENT4[GLD-4], demonstrating that increased paralysis in TENT4[GLD-4]-loss worms is mediated by $\beta$-amyloid[1-42] (Fig 1A). The toxicity of TENT4[GLD-4] depletion was partially rescued by O4, a small molecule that reduces $\beta$-amyloid[1-42] toxicity [33], suggesting that paralysis was, at least in part, due to an increase in $\beta$-amyloid[1-42] conformational toxicity (Fig 1C). The protective effect of TENT4[GLD-4] was not limited to $\beta$-amyloid[1-42], as depletion of TENT4[GLD-4] moderately suppressed aggregation and increased toxicity of $\alpha$-synuclein-YFP in the *C. elegans* model of Parkinson's disease, NL5901 [34] (Fig 1D–H). Although increased variability of the data exists, these results indicate that TENT4[GLD-4] protects against amyloid-induced toxicity from two distinct amyloidogenic peptides. We chose to expand our investigation to *TENT2[GLD-2]* as this noncanonical TENT, along with *TENT4b[GLD-4]*, is described as a major poly(A)polymerase in *C. elegans* [30, 32]. RNAi depletion of TENT2[GLD-2] also resulted in increased paralysis (Fig S1A and C) and the accumulation of toxic $\beta$-amyloid[1-42] species in CL2006 *C. elegans* (Fig 2B) even if it decreased $\beta$-amyloid[1-42] transgene levels (Fig S1B). As observed with TENT4[GLD-4], increased $\beta$-amyloid[1-42] toxicity in TENT2[GLD-2]-deficient CL2006 was rescued by treatment with O4 (Fig S1D). In addition, TENT2[GLD-2]-loss worms showed increased $\alpha$-synuclein-YFP toxicity as measured by body bends per minute (Figs S1E and F and S2A). This was associated with an increase in $\alpha$-synuclein-YFP aggregates suggesting a different protective activity of *TENT2[gld-2]* for $\alpha$-synuclein-YFP (Fig S2B–E). TENT2[GLD-2] loss had no effect on body bends in nontransgenic worms (N2) indicating the observed toxicity was dependent on $\alpha$-synuclein-YFP expression and not TENT2[GLD-2] loss (Fig S1F). Taken together, these results indicate that TENTs facilitate the formation of nontoxic amyloidogenic assemblies.

## The RNA exonuclease, Exosc10, potentiates amyloid toxicity

TENT4b-driven physiological amyloidogenesis is opposed by the RNA exosome [26]. We focused our investigation on the conserved exonuclease Exosc10, which robustly inhibits intracellular amyloidogenesis in different subcellular compartments of mammalian cells [26]. Depletion of the ExoSC10 homolog CRN-3 reduced $\beta$-amyloid[1-42] toxicity in the *C. elegans* model of Alzheimer's disease (Figs 2A and S1A). In fact, age-associated delays in $\beta$-amyloid[1-42] proteotoxicity in RNA exosome–loss worms were as profound as those observed by inhibition of the longevity regulator insulin/IGF receptor, Daf-2 [28], during early disease progression. The depletion of CRN-3 resulted in a partial rescue of GLD-4 and GLD-2–induced paralysis phenotype while suppressing levels of toxic $\beta$-amyloid[1-42] that accumulates in TENT4[GLD-4]- or TENT2[GLD-2]-depleted worms (Figs 2A and B and S1A and C). EXOSC10[CRN-3] depletion did not affect transgene $\beta$-amyloid[1-42] mRNA levels (Fig S1B). As with the TENTs, these effects were not limited to $\beta$-amyloid[1-42] because depletion of EXOSC10[CRN-3] increased $\alpha$-synuclein aggregation in NL5901 worms (Figs 2C–F and S1E). EXOSC10[CRN-3]-loss worms displayed large apparent aggregates with an increased percentage of aggregate area per worm (Fig 2C–F). These large aggregates were partially dependent on TENT4[GLD-4] and TENT2[GLD-2] activity, suggesting that aggregation of $\alpha$-synuclein is modulated by the RNA tailing and RNA exosome machinery (Figs 2C–F and S2F–I). As with $\beta$-amyloid[1-42], loss of TENT4[GLD-4]-induced $\alpha$-synuclein toxicity, as measured by body bends per minute, was rescued by codepletion of EXOSC10[CRN-3] (Fig 2G). Codepletion of EXOSC10[CRN-3] did not significantly rescue TENT2[GLD-2]-induced $\alpha$-synuclein toxicity, likely because of the robust phenotype of TENT2[GLD-2] as compared to TENT4[GLD-4] (Fig S2A and J). As observed with TENT4[GLD-4] or TENT2[GLD-2] depleted N2 worms, neither EXOSC10[CRN-3] loss nor EXOSC10[CRN-3]/TENT codepletions had any effect on body bends in nontransgenic worms (N2), indicating the observed toxicity was dependent on $\alpha$-synuclein-YFP expression (Fig S1F). Taken together, these results suggest EXOSC10[CRN-3] potentiates amyloid toxicity.

## RNA exosome loss delays cognitive impairment and restores memory in AD mouse models

Based on results obtained in *C. elegans*, we asked whether ExoSC10 depletion could protect against amyloid toxicity in mouse models of Alzheimer's disease. The 5xFAD (familial AD) transgenic mice are widely used to study $\beta$-amyloid[1-42] toxicity in the brain [35]. These mice accumulate high levels of intraneuronal $\beta$-amyloid[1-42] beginning around 2 mo of age and exhibit many AD-related phenotypes including the appearance of $\beta$-amyloid[1-42] plaques and cognitive impairment measurable within 3 mo. Intranasal delivery of siRNA efficiently depleted endogenous ExoSC10 without affecting $\beta$-amyloid[1-42] protein levels or gross locomotor activity in 5XFAD mice (Figs 3A–C and S3A and B). This provided a paradigm to test the effect of RNA exosome depletion on $\beta$-amyloid[1-42] age-associated toxicity in the context of the mouse brain in vivo. We next investigated whether depletion of ExoSC10 rescues cognitive impairments in 5xFAD mice. Mice display innate preference for novel objects [36]. In the novel object

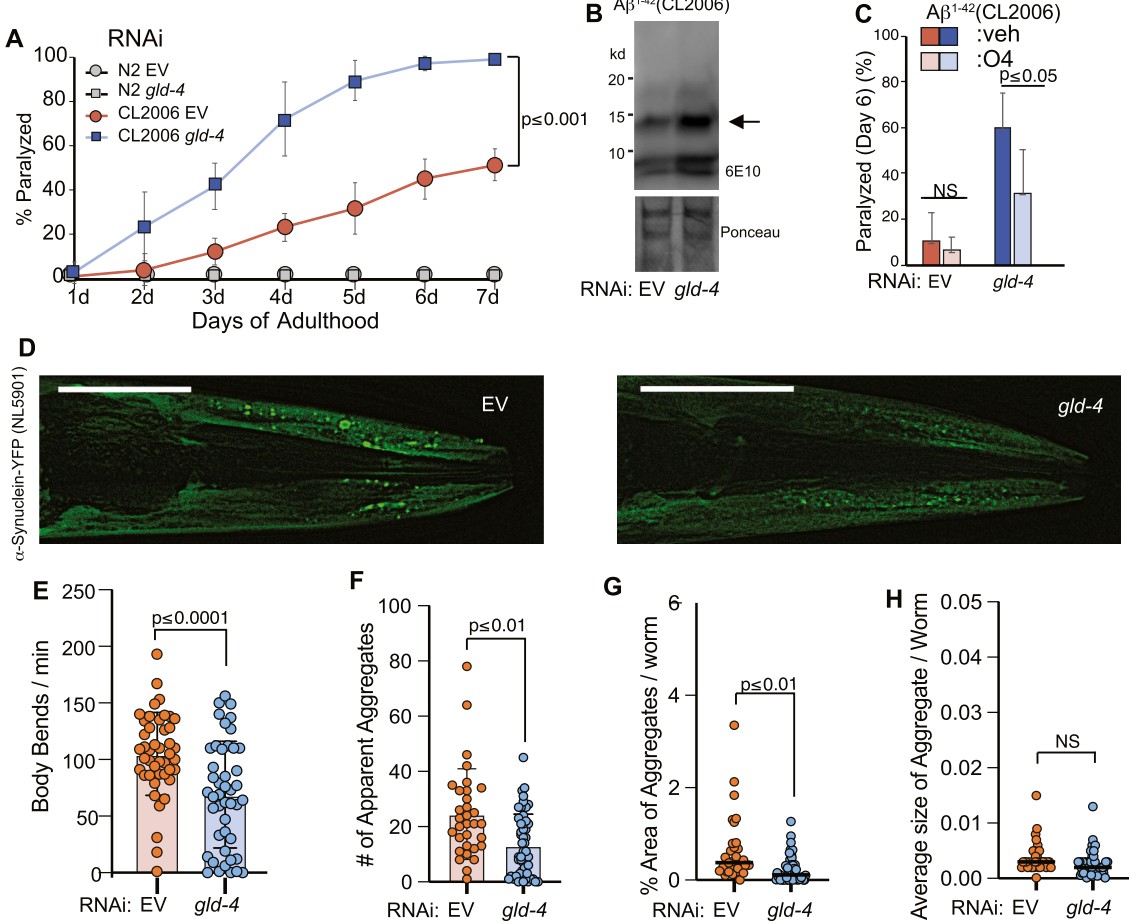

**Figure 1. RNA tailing machinery protects against amyloid proteotoxicity in Alzheimer's and Parkinson's disease *C. elegans* models.**
**(A)** TENT4[GLD-4] knockdown accelerates paralysis in the β-amyloid–expressing CL2006 strain but does not affect motility in nontransgenic N2 worms. Paralysis was measured as % worms paralyzed over days of adulthood (n = 3). **(B)** TENT4[GLD-4] suppresses soluble oligomer formation of β-amyloid. SDS–PAGE Western blots showing soluble β-amyloid (6E10 antibody) in CL2006 after RNAi against TENT4[GLD-4]. The arrow indicates soluble multimeric species. Ponceau staining is shown as a loading control. The arrow indicates soluble multimeric species believed to be the toxic form of β-amyloid. Ponceau is shown as a loading control. **(C)** Small molecule O4 rescues paralysis induced by TENT4[GLD-4] depletion (n = 3). **(D)** Representative fluorescence images of α-synuclein-YFP aggregates in day 7 adult NL5901 worms after RNAi knockdown of TENT4[GLD-4] or empty vector (EV) control. n = 32 EV, 50 TENT4[GLD-4]. Scale bar = 100 μm. **(E)** Measurement of body bends per minute in NL5901 day 7 adult worms shows depletion of TENT4b[GLD-4] promotes α-synuclein-YFP toxicity. EV, empty vector negative control. n = 54 EV, 53 TENT4b[GLD-4]. **(F, G, H)** Quantification of apparent α-synuclein-YFP aggregates in day 7 adult NL5901 *C. elegans* strains fed with indicated RNAi. **(F)** Number of apparent α-synuclein-YFP aggregates per worm. **(G)** Quantification of the percent area of α-synuclein-YFP aggregates per worm. **(H)** Quantification of α-synuclein-YFP aggregate size per worm. Statistical *P*-values are from a two-tailed *t* test.
Source data are available for this figure.

recognition assay, mice are tested to determine whether they prefer a novel object (N) to a familiar (F) object that they have previously encountered. Consistent with the literature, 3-mo-old 5xFAD mice treated with scramble siRNA were impaired, as they spent similar time between the novel and familiar objects (Fig 3D). In contrast, 5xFAD mice treated with siRNA targeting ExoSC10 spent more time with the novel object, suggesting that RNA exosome loss delays cognitive impairment in the 5xFAD model (Fig 3D). The two-trial Y-maze assay evaluates spatial working memory by determining whether mice remember the arm of the maze they did not explore before (37). Akin to the novel object recognition test, 5xFAD mice treated with scramble siRNA exhibit spatial memory impairment as they spend equivalent amount of time in the familiar and novel arms, whereas 5xFAD mice treated with siRNA targeting

ExoSC10 spent significantly more time in the novel arm (Fig 3E). These results implied that ExoSC10 depletion delays the spatial working memory impairment of 5xFAD mice. Although we do not observe obvious differences in plaque loads between control and Exosc10-depleted mice (Figs 3A–C and S3C and D), there were significant differences in the levels of A11 conformation-specific antibody staining (38, 39) (Fig 3F and G). Next, we tested whether ExoSC10 depletion could restore memory of 20-mo-old 3xTg AD mice that become cognitively impaired at 6 mo of age (40, 41, 42). Alike 5xFAD mice, intranasal injection of Exosc10 siRNA for 1 wk did not affect locomotor activity in these mice (Fig S3E and F) and was associated with a decrease in A11 staining (Figs 3H and S3G and H). Remarkably, ExoSC10 depletion restored novel object recognition and spatial memory in 20-mo-old 3xTg-AD mice, indicating that

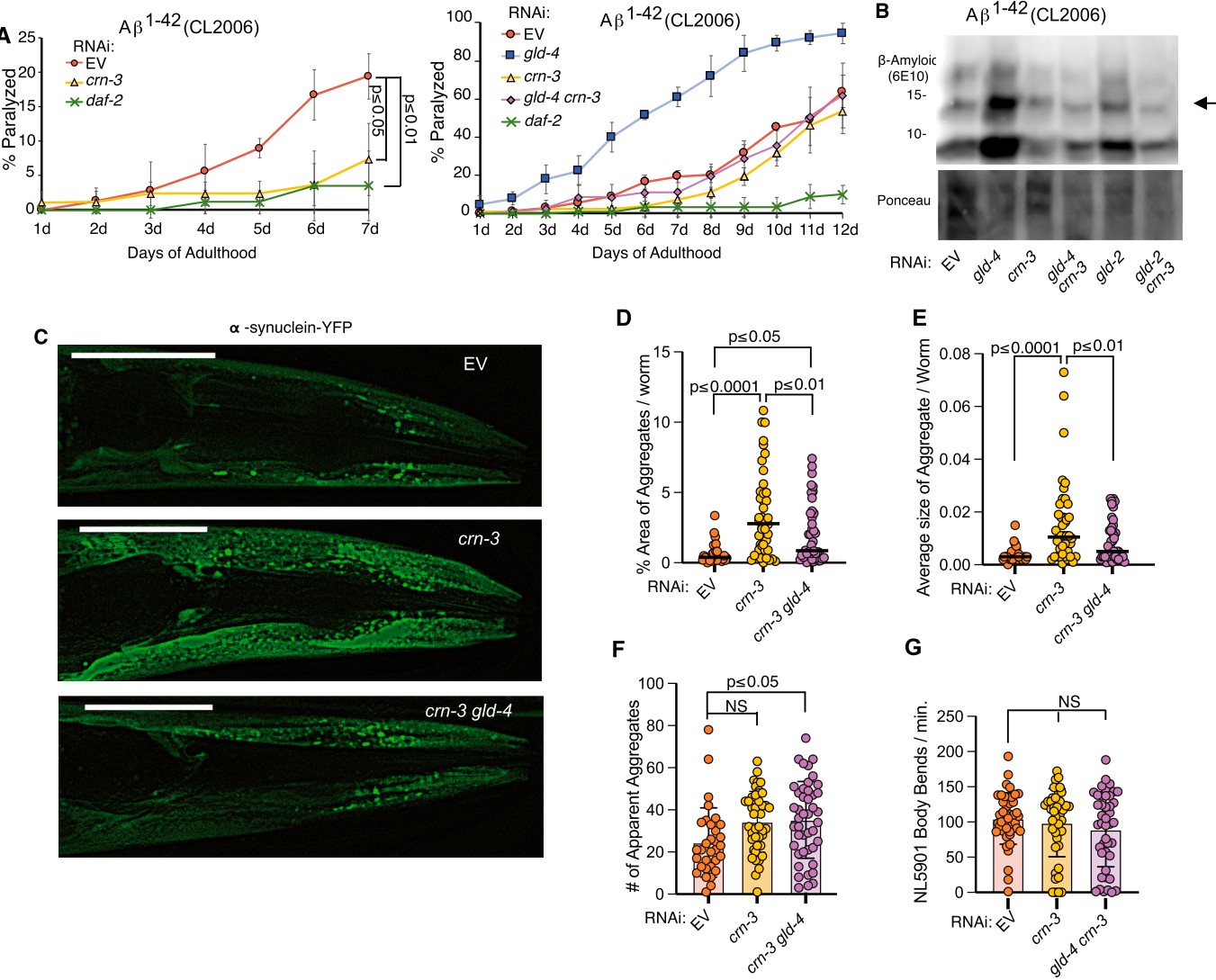

**Figure 2. RNA exosome promotes amyloid toxicity in Alzheimer's and Parkinson's disease *C. elegans* models.**
**(A)** Paralysis curves of CL2006 worms fed RNAi targeting indicated genes (n = 3). **(B)** SDS–PAGE Western blots showing soluble β-amyloid (6E10 antibody) in CL2006 after RNAi against TENT4[GLD-4], TENT2[GLD-2], and ExoSC10[CRN-3]. The arrow indicates soluble multimeric species. Ponceau staining is shown as a loading control. **(C)** Representative fluorescence images of α-synuclein-YFP aggregates in day 7 adult NL5901 worms after RNAi treatments; EV, empty vector control. Scale bar = 100 μm. **(D, E, F)** Quantification of apparent α-synuclein-YFP aggregates in day 7 adult NL5901 *C. elegans* strains fed indicated RNAi. EV dataset is the same as Fig 1D, F–H. n = 32 EV, 44 ExoSC10[CRN-3], 44 ExoSC10[CRN-3] TENT4[GLD-4] worms. **(D)** Quantification of the percent area of α-synuclein-YFP aggregates per worm. **(E)** Quantification of α-synuclein-YFP aggregate size per worm. **(F)** Number of apparent α-synuclein-YFP aggregates per worm. **(G)** Measurement of body bends per minute in NL5901 day 7 adult worms fed with indicated RNAi. The EV dataset is the same as presented in Fig 1E. EV, empty vector negative control. n = 54 EV, 43 ExoSC10[CRN-3], 44 TENT4[GLD-4] ExoSC10[CRN-3]. **(A, D, E, F, G)** Statistical *P*-values are from a two-tailed *t* test (A) and ANOVA (D, E, F, G).
Source data are available for this figure.

learning and memory can be reestablished by ExoSC10 loss (Fig 3I and J). Taken together, these results demonstrate that RNA exosome activity potentiates pathological amyloid proteotoxicity in both *C. elegans* and mouse models of Alzheimer's disease.

Together, our findings reveal that enzymes involved in physiological amyloidogenesis are also implicated in pathological amyloid toxicity. This framework offers a new lens to examine whether proteopathic conditions can arise in part from dysregulation of physiological amyloid control mechanisms: a concept with potential for therapeutic targeting. Although we cannot

formally conclude that TENT4b/2 activity or Exosc10 loss reduces β-amyloid[1-42] toxicity through conformational changes in *C. elegans* and mouse models, we note that this correlation is consistent with reports of oligomeric toxicity of β-amyloid[1-42] (32, 38, 43). TENT2[GLD-2] loss–associated α-synuclein aggregation suggests the involvement of alternative toxicity mechanisms, including the formation of aggregates more prone to shedding (43, 44), a phenomenon that remains challenging to definitively test in *C. elegans*. In addition, the exact mechanisms by which the RNA tailing and exosome axis modulates pathological amyloid toxicity

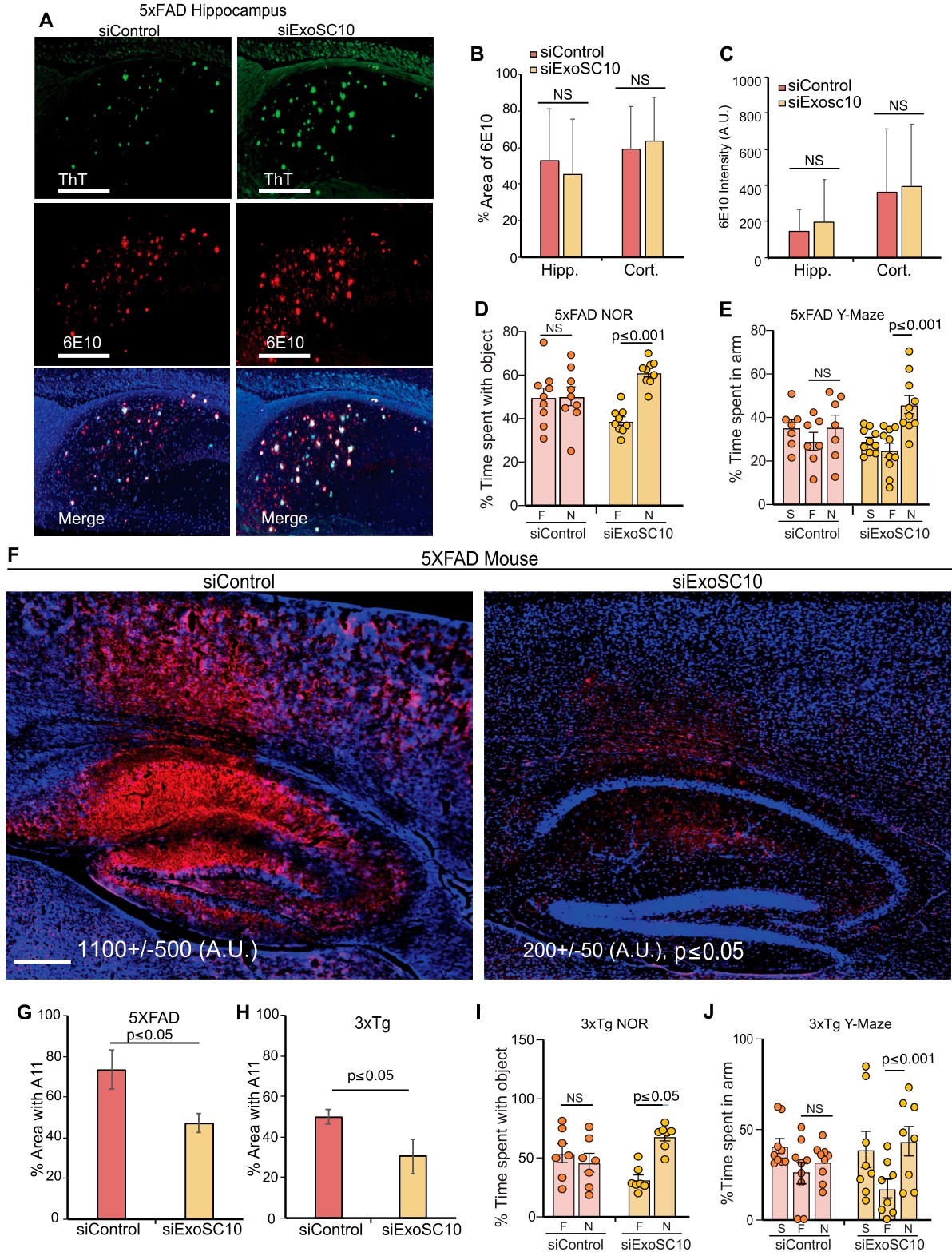

**Figure 3. Inhibition of RNA exosome rescues cognitive impairment in Alzheimer's disease mice.**
**(A)** Representative images of 5xFAD mouse hippocampus treated with control (siControl) or ExoSC10 targeting siRNA (siExoSC10) stained for thioflavin (ThT), $\beta$-amyloid (6E10), and DAPI. Scale bar = 100 $\mu$m. **(B)** Quantification of percent area of $\beta$-amyloid (6E10) staining in the hippocampus and cortex of siControl or siExoSC10-treated 5xFAD mice. n = 5, NS, not significant. **(C)** Quantification of $\beta$-amyloid (6E10) intensity in the hippocampus and cortex of siControl or siEcoSC10-treated 5xFAD mice. n = 5, NS, not significant. **(D, E)** Behavioral tests in 3-mo-old 5xFAD mice show Exosc10 depletion delays impairment in novel object recognition ((D); % time spent with familiar [F] or novel [N] object; n = 9) and spatial working memory ((E); % time exploring start [S], familiar [F], or novel [N] arms; n = 7 siControl, 10 siExoSC10; NS, not significant). **(F, G)** Immunostaining for oligomeric species (A11) in 5xFAD hippocampus (F) and quantification of A11-positive area ((G); n = 3). **(F)** Fluorescence intensity

remain an open question. Nonetheless, the protective effects of EXOSC10 depletion observed in Alzheimer's disease models highlight new avenues of investigation in proteopathies, link pathways of physiological amyloidogenesis to pathological amyloids, and identify pharmacological inhibition of EXOSC10 as a compelling therapeutic avenue.

# Materials and Methods

## Drug treatments

For *C. elegans* experiments, 2,8-BIS(2,4-dihydroxyphenyl)-7-hydroxy-3H-phenoxazin-3-one (O4, TMT00595; Sigma-Aldrich) was incorporated into agar at 100 $\mu$M before bacterial seeding.

## Animal models

5xFAD mice (B6.Cg-Tg(APPSwFlLon,PSEN1M146LL286V)6799Vas/Mmjax, 8–16 wk) were purchased from Jackson Labs/MMRRC.

3xTg mice (B6;129-Psen1tm1MpmTg(APPSwe,tauP301L)1Lfa/Mmjax, 17–22 mo) were bred in-house. All mice were maintained under standard conditions, with protocols approved by the University of Miami IACUC.

## *Caenorhabditis elegans* model organisms

Strains CL2006 (expressing A$\beta$1–42), NL5901 ($\alpha$-synuclein-YFP), and N2 wild type were from the Caenorhabditis Genetics Center. Worms were cultured at 20°C. RNAi clones for crn-3 (Horizon Discovery), gld-4, and daf-2 were cloned into pL4440 and transformed into HT115 bacteria for feeding assays.

## Immunofluorescence and amyloid staining

For mouse brain sections, immediately after behavioral testing, mice were anesthetized and transcardially perfused with PBS. Brains were postfixed with 4% PFA (#P614; Sigma-Aldrich), cryoprotected in 30% sucrose snap-frozen in OCT (O.C.T., Tissue-Tek), and stored at –80°C. Antigen retrieval was done by incubating slides for 30 min. in boiling sodium citrate antigen retrieval buffer (0.1 M sodium citrate tribasic dihydrate, 0.1% Tween, pH = 6). Slides were washed in PBS, then in gelatin/Triton permeabilization buffer (0.2% weight-by-volume gelatin, 0.25% weight-by-volume Triton X-100). For A11, after postfix, sections were washed in PBS and then treated with 70% formic acid for 15 min at room temperature. Slides were blocked in 10% normal goat serum in PBS (50062Z; Thermo Fisher Scientific) for 1 h. APP-6E10 (NBP2-62566; Novus) was diluted 1:100 in 5% goat serum in PBS and incubated overnight at RT, then washed in PBST, and incubated with goat anti-rabbit Alexa 594–conjugated

secondary antibody (1:500A-11008; Life Technologies). Slides were washed in PBST on a rocker and incubated with Hoechst 33258 (1:1,000; Thermo Fisher Scientific) or thioflavin S (0.01% in 50% ethanol), then washed in 50% ethanol. Slides were then washed in PBS and mounted in 80% glycerol or Vectashield Vibrance Antifade Mounting Media (H-1700-2; Vector labs). Images were obtained using a Keyence BZ-X800 microscope with a 20X objective (Nikon CFI S Plan Fluor ELWD 20X/0.45NA), which were subsequently stitched together using Keyence BZ-X800 Analyzer software. All images were equally black-balanced and deconvolved using Keyence BZ-X800 Analyzer software.

NL5901 worms were immobilized with 25 mM NaN3 on agar pads and imaged using a 40X objective (Nikon CFI S Plan Fluor ELWD ADM 40XC/0.6NA). Aggregates were quantified using Fiji by thresholding and particle analysis.

## Immunoblot

For mice, hippocampus and cerebral cortex were rapidly dissected in ice-cold phosphate-buffered saline, snap-frozen, and stored at –80°C before use. Brain regions were homogenized in ice-cold Triton lysis buffer 20 mM Tris–HCl, pH 7.4, 150 mM NaCl, 1 mM EDTA, 1 mM EGTA, 1% Triton X-100, 10 $\mu$g/ml leupeptin, 10 $\mu$g/ml aprotinin, 5 $\mu$g/ml pepstatin, 1 mM phenylmethanesulfonyl fluoride, 1 mM sodium vanadate, 50 mM sodium fluoride, and 100 nM okadaic acid. The lysates were centrifuged at 15,000$g$ for 10 min to remove insoluble debris, and protein concentrations in the supernatants were determined using the Bradford protein assay. Proteins (10–20 $\mu$g) in brain region extracts were resolved with SDS–PAGE, transferred to nitrocellulose membranes, and immunoblotted with primary antibodies. For CL2006, worms were synchronized and plated onto appropriate RNAi plates for two generations. ~200–500 day 3–old adults were collected, lysed in buffer containing 50 mM Tris–HCl, pH 8.0, 0.5 M NaCl, 4 mM EDTA, 1% NP-40 using sonication (50% power for 10 s, six times), and spun at 14,000G for 10 min. An equal concentration of protein was loaded for Western blots and was probed with APP-6E10 (NBP2-62566; Novus). Age is defined as days past the L4 stage. For all, primary antibodies used (1:1,000) were the following: APP-6E10 (NBP2-62566; Novus), Exosc10 (sc-374595; Santa Cruz), beta-actin (MA5-15739; Life Tech). For 6e10 blots, the PVDF membrane was submerged in boiling PBS for 5 min before incubating with blocking buffer.

## qRT-PCR

RNA samples were DNase-treated and reverse-transcribed with the High Capacity cDNA RT kit. qRT-PCR was performed with PowerUp SYBR Green Master Mix on an ABI StepOne system. Primers are listed in Table S1.

values shown in (F) (arbitrary units). Merged image of DAPI and red channels. Scale bar = 200 $\mu$m. **(H)** Percent area of A11 staining in 3xTG mouse hippocampal and cortical field of view as shown in Fig S3G. Related to Fig S3G and H (n = 3). **(I, J)** Exosc10 depletion rescues novel object recognition (I) and spatial working memory impairments (J) in 20-mo-old 3xTg-AD mice (n = 7 and n = 9 siControl, 8 siExoSC10, respectively; NS, not significant). Error bars represent ± SEM. **(B, C, D, F, G, H, I, J)** Statistical tests: two-tailed *t* test (B, C, F, G, H, I) and ANOVA (D, E, I, J).

### C. elegans behavioral assays

CL2006 paralysis was scored daily post-L4 stage on RNAi plates, with at least three independent replicates. NL5901 worms were assessed at day 7 for body bends over 60 s in M9 buffer by counting head/tail bends returning to midline.

### Mouse depletion treatments

Mice were injected intranasally with 10 $\mu$g of scramble siRNA (#5, AM4642; Ambion) or ExoSC10 siRNA (J-049286-05; Dharmacon) in PBS, and 5 $\mu$l was administered in each nostril every other day for a week consistent with the protocol described in references (46) and the behavior was initiated on the third day (open field) and on the sixth day (Y-maze and novel object recognition).

### Mouse behavioral assay

#### Open field
The locomotor activity in an open field was measured as previously described (45). Briefly, mice were placed in a Plexiglas open field (San Diego Instrument) outfitted with photobeam detectors under soft overhead lighting, and activity was monitored during 30 min using activity monitoring software (San Diego Instrument).

#### Novel object recognition (NOR)
Two identical copies of object 1 were placed in a Plexiglas arena (50 cm long × 20 cm wide × 25 cm tall) equidistant from the walls of the container. Mice were individually placed within the arena and were allowed to explore the objects during a 5-min habituation phase. After this, mice were removed from the arena and placed in a holding container for 5 min. Mice were reintroduced to the same arena and presented with one copy of object 1, as well as a novel, never previously encountered object 2. The total time each mouse spent exploring object 1 (familiar, F) compared with object 2 (novel, N) was recorded (exploration was defined as touching the object with the nose or forelimbs, sniffing the object, or closely approaching the object in a forward, attention-directed lunge) as we previously reported (46). Both the percentages of time spent with each object were reported.

#### Two-trial Y-maze
The Y-maze apparatus (Stoelting Co.) (lane width 5 cm × arm length 35 cm × arm height 10 cm) was arranged on an elevated platform with one arm facing outward toward the experimenter (start arm, S) and two distal arms facing inward toward the back wall of the room. Extra-maze visual cues of varying shapes and sizes were placed on the walls of the room ranging from 0.5- to 2-m distance from the maze itself. During a preliminary acquisition phase, the left or the right distal arm was obstructed by placing a barrier block at the entrance of the arm. Mice were individually placed into the end of the start arm and allowed to explore the open areas of the maze for 5 min. After this, mice were removed from the maze and returned to their home cage during a 30-min intertrial interval (ITI). During a 2-min retrieval trial, mice were returned to the start arm and allowed to freely explore all arms, including the previously obstructed, novel arm (N). Measurement of exploratory behavior

began when a mouse had left the start arm, and the total time each mouse spent exploring the novel arm compared with the familiar arm was calculated. An arm entry was defined as placing all four paws within the arm, and a mouse was considered to have exited an arm when all four paws were located outside of the arm. Periods in which a mouse engaged in self-grooming or remained stationary were excluded from the final calculation. Mice were assessed using a 1-min ITI between acquisition and retrieval trials to confirm preference for novelty and sufficient visual acuity to recognize extra-maze visual cues, and to control for potential anxiety-like motivational disturbances that may have influenced exploratory behavior.

### Statistical analysis

Quantitation of microscopy-based data was performed using Fiji on at least three representative images. Graphs represent mean values, and error bars depicted represent SD between repeats, unless otherwise stated in the figure legend. Appropriate statistical analyses were performed (e.g., two-tailed $t$ test, or two-way analysis of variance [ANOVA] for multiple comparisons with Tukey's post hoc test) using Excel or Prism software.

## Data Availability

All unique/stable reagents generated in this study are available from the corresponding author with a completed materials transfer agreement. All other data are included in the article and/or SI Appendix.

## Supplementary Information

## Acknowledgements

We thank Ms. Carolie Crawley for technical expertise. This work was supported by NIH grants from the National Cancer Institute (R01CA275828) to S Lee, National Institute of General Medical Sciences (R35GM149221, R01GM115342) to S Lee, Sylvester Comprehensive Cancer Center to S Lee, National Institute of Aging (K99AG080474) to M Bokros, and the National Institute of Mental Health (R01MH104656) to E Beurel.

### Author Contributions

M Bokros: conceptualization, data curation, formal analysis, funding acquisition, investigation, visualization, methodology, and writing—original draft, review, and editing.
A Grunfeld: formal analysis, validation, investigation, methodology, and writing—review and editing.
NC Balukoff: conceptualization, formal analysis, validation, investigation, and methodology.

J Bouviere: validation, investigation, and writing—original draft, review, and editing.

E Beurel: conceptualization, resources, data curation, formal analysis, supervision, funding acquisition, validation, investigation, visualization, methodology, project administration, and writing—original draft, review, and editing.

S Lee: conceptualization, resources, data curation, formal analysis, supervision, funding acquisition, validation, investigation, visualization, methodology, project administration, and writing—original draft, review, and editing.

## Conflict of Interest Statement

The authors declare that they have no conflict of interest.

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
