## [Reviewer comments · Life Science Alliance]

Enzymes of Physiological Amyloidogenesis Control Pathological Amyloid Toxicity

Michael Bokros, Alex Grunfeld, Nathan Balukoff, Jessica Bouviere, Eleonore Beurel, and Stephen Lee
DOI: <https://doi.org/10.26508/lsa.202503493>

Corresponding author(s): Stephen Lee, University of Miami

Review Timeline:	Submission Date:	2025-08-21
	Editorial Decision:	2025-10-16
	Revision Received:	2026-01-28
	Editorial Decision:	2026-02-20
	Revision Received:	2026-02-27
	Accepted:	2026-03-03

Scientific Editor: Tim Fessenden

Transaction Report:

October 16, 2025

Re: Life Science Alliance manuscript #LSA-2025-03493-T

Prof. Stephen LEE
University of Miami
Biochemistry and Molecular Biology
1011 NM 15th Street, Room 217
Miami, Florida 33136

Dear Dr. LEE,

Thank you for submitting your manuscript entitled "Enzymes of Physiological Amyloidogenesis Control Pathological Amyloid Toxicity" to Life Science Alliance. The manuscript was assessed by expert reviewers, whose comments are appended to this letter.

As you will see, all reviewers commended these findings on RNA tailing enzymes and neurotoxic amyloids although they differed somewhat in their enthusiasm. All made important requests that should be resolved prior to publication, with changes to the text or with new data. In particular Reviewer 1 remarked that the proposed mechanism of formation of non-toxic amyloids requires additional data for support and suggests several avenues to approach this. This reviewer also requests observations on Abeta to accompany those already provided using alpha-synuclein. Reviewer 3 requests validation of knockdown efficacy. If possible, their request to include results from a neuronal expression model in worms should also be addressed. Finally, Reviewer 3 made several important suggestions to clarify the proposed model and appropriately contextualize these results by changes to the text. Additional data not mentioned here are not required in a revised manuscript.

Thank you for this interesting contribution to Life Science Alliance. We are looking forward to receiving your revised manuscript.

Sincerely,

-- Summary blurb (enter in submission system): A short text summarizing in a single sentence the study (max. 200 characters

including spaces). This text is used in conjunction with the titles of papers, hence should be informative and complementary to the title and running title. It should describe the context and significance of the findings for a general readership; it should be written in the present tense and refer to the work in the third person. Author names should not be mentioned.

B. MANUSCRIPT ORGANIZATION AND FORMATTING:

Reviewer #1 (Comments to the Authors (Required)):

Summary

This is a very interesting paper with an important research question - the authors investigate whether RNA tailing dynamics, which they have previously shown modulate the maturation of aggregates, is important for modulating toxicity of human disease related proteins A β and α -syn. This paper very effectively shows that wild type function of TENT4, and inhibition of the opposed RNA exosome each reduce toxicity of these disease-associated proteins. In particular, the mouse data in figure 3 is very striking and shows a strong removal of oligomeric A β in a mouse model, along with associated improvements in memory. The results described in this paper are of great interest to the field. With that said, I have concerns that the authors have not proven a key part of the mechanism which they propose, and that more experimental work will be required to enable them to claim this.

Major:

The claim that wild-type action of TENTs and inhibition of crn-3 mitigate A β / α -syn toxicity via formation of non-toxic amyloids is poorly supported.

- In the case of α -synuclein in worms, some evidence of aggregation is provided in the form of confocal microscopy and analysis of aggregate number and sizes. However, no evidence is provided that A β behaves similarly. Indeed, when amyloid plaque size is assessed in mice in Fig 3, there is no difference in plaque size. Figure 1C shows a partial rescue of gld-4 and gld-2 by O4 which drives fibrillation and suppresses toxicity, however this is an orthogonal experiment which demonstrates that reducing toxic species via O4 driven fibrillation mitigates toxicity but doesn't otherwise demonstrate the mechanism of action discussed. Other interpretations such as removal of toxic oligomers would also explain the observations provided for TENTs and RNA exosome. Furthermore, the α -syn puncta may not represent amyloids (in our experience these are often soluble), or even insoluble proteins and this needs to be further explored. Therefore, while the authors propose that formation of non-toxic amyloids underlies the reduction in toxicity, there has not been a sufficient investigation of the formation of these amyloids in the context A β and α -syn.

- o The authors should quantify the soluble/insoluble fraction of the A β and α -syn worms in the various treatments or otherwise conduct more robust investigations of this point.

- o The authors should use X-34 staining or a similar technique to identify the amyloids which are central to the model they propose.

- o More clearly show how total A β levels change relative to oligomeric species and amyloids (quantify gels, etc)

- It is observed that inhibition of gld-4 and gld-2 both increase paralysis. It appears that gld-4 inhibition decreases "aggregation" of α -syn, in line with the papers conclusion that TENTs facilitate the formation of non-toxic amyloid species. However, gld-2 RNAi displays the opposite action on α -syn aggregation, suggesting that its wild type action opposes the formation of aggregates, appearing to contradict the conclusions that TENTs in general facilitate the formation of non-toxic amyloid aggregates, both because GLD-2 does not appear to do this and because in the gld-2 RNAi context α -syn aggregates appear to be associated with higher toxicity.

- o gld-2 should be moved to the supplement or more thoroughly investigated.

- o Clearer discussion of gld-2 in general in relation to the different mechanism.

Minor

Intro - Previous studies have shown that TENT4b facilitates formation of mature amyloidogenic aggregates, however this seems to only have been shown in nuclear localised A-bodies, and A β and α -syn mostly aggregate elsewhere. Some discussion of the

potential mechanism by which RNA tailing of long-noncoding RNA derived from the ribosomal intergenic spacers would influence disease protein aggregation in the cytoplasm should exist in the text.

Line 112 - *gld-2* & *gld-4* inhibition is said not to change A β transgene expression, but *gld-2* significantly decreases it. This should be mentioned in the text.

Figure 2 D,E,F,G: Should show *gld-2*, *gld-4* controls as well as doubles with *crn-3*

Fig S1C - A *crn-3* alone control would be helpful here

Figures general - Clearer labelling of gels which show monomer/soluble/oligomeric A β to show what the bands are said to be.

Figures general - Quantification of western blots required throughout

Figure S1B legend - show the n= for the qPCR.

Fig 3 panels should be rearranged to match with the order they appear in the text

Line 463 references Amylo-Glo staining, but this is not shown in the paper.

More detail on the methods for brain & worm lysates should be shown in methods section.

More detail on intranasal administration of the treatment in mice, including buffers used & how long between end of regime and behavioural testing.

Misc

72 Raises the question if -> raises the question of whether

73 modules -> modulates

104: confirmation -> conformational

129: *Daf-2* -> DAF-2

Unless there are restrictions by the journal on figure size, please move data S2g, h into figure 3. It would be helpful to also include the model from S1 in the main figures. Could the size of the depicted aggregates in the model be kept the same.

S2A, E: are these whole brains that have been analyzed? Please clarify in legend.

S2D has an error bar which is out of position

Provide source data for all quantifications in the figures.

Reviewer #2 (Comments to the Authors (Required)):

Review of "Enzymes of Physiological Amyloidogenesis Control Pathological Amyloid Toxicity"

Authors: Michael Bokros, Alex Grunfeld, Nathan C. Balukoff, Jessica Bouviere, Eléonore Beurel, and Stephen Lee

Summary: This paper uses *C. elegans* and mouse models to demonstrate that manipulating the RNA tailing/decay axis alters the toxicity of β -amyloid and α -synuclein. Their data indicate that the RNA tailing enzymes, TENT4b and TENT2 protect against β -amyloid and α -synuclein-induced toxicity by favoring formation of amyloidogenic assemblies, which they consider "non-toxic". On the other hand, the RNA exonuclease, Exosc10, increases toxicity associated with the accumulation of amyloidogenic intermediates, presumably soluble oligomers. Exosc10 depletion prevents cognitive decline and restores memory in two mouse models.

Comments:

1. This paper largely confirms a consensus opinion in the amyloid field that the main toxic species in protein aggregation diseases are soluble oligomers, while insoluble fibrils are less toxic. Their data are generally consistent with and confirm this opinion. They tend to overstate or state too starkly the difference in toxicity of oligomers and fibrils, however. It's not quite as all-or-nothing as they suggest. In some studies, fibrils do have some toxicity.
2. Similarly, their work tends to confirm a growing consensus linking RNA metabolism to amyloid formation. Thus, TENT4b and TENT2 tend to abrogate toxicity, while exonucleases increase it. However, while their data are generally valid and support their claims, some of the details of their studies are more ambiguous than they acknowledge. In Figure 1, for example, the difference between EV and *gld4* (panel E and even more so panel F) is statistically significant but there is a lot of overlap in the two groups. In panel H, the difference between *gld2* and either EV or *gld4* seems to depend entirely on two outlier points in *gld2* worms. Similar comments apply to some of the data shown in Figure 2. A more nuanced approach to such data is warranted. While the statistical significance exists, they could be more forthcoming about limitations in their data.
3. What kind of gel is shown in Figure 2B (and elsewhere)? This should be commented upon.
4. There are many small errors in the language. They should do a thorough proofreading before resubmitting their manuscript.
5. The main limitation of this paper is that while it expands some of the phenomenology of amyloid-related diseases, it really does not get into mechanism at all. Exactly how might RNA alter the course of these diseases. Do they mean any and all RNA? Why enzymes affecting RNA tailing in particular? Is the effect of RNA direct (e.g., by binding β -amyloid or α -synuclein) or indirect? If the former, how generalizable is the finding? If the latter, what other players are involved.

Reviewer #3 (Comments to the Authors (Required)):

Summary: This manuscript conducts preliminary RNAi screens in *C. elegans* examining mRNA tailing and degradation.

Subsequent efforts in mice are to translate the findings. This work is potentially impactful in its identification of modifiers of amyloid and synuclein toxicity. The translational effort also seems promising. However, the paper exhibits limitations in the rigor of the methods and therefore generalizability of these findings. In particular, using true genetic loss of function would enhance

rigor. Likewise, increasing the powering particular of rodent behavior would be helpful.

- This study is heavily dependent upon RNAi methodology and body wall muscle expression of amyloid which has been shown to not necessarily translate well to neurodegeneration in AD.
- The specificity of RNAi here is not demonstrated for any of the genes targeted either in *C. elegans* or mice. For example, does crn-3 RNAi treatment impact crn-3 transcript levels and/or protein abundance.
- Including validation of at least the main findings using a neuronal amyloid model would be more disease relevant and increase the impact of this work. An ideal experiment would be to recapitulate the RNAi findings from muscle instead in neurons using genetic ablation of the genes in question. This could be accomplished using existing crn-3 conditional or partial loss of function alleles.
- While the number of animals studied for histology seems marginally sufficient, the behavioral studies seem underpowered, especially given the lack of validation and measurement of siRNA treatment in rodents.

General Comments to Reviewers

We would like to express our sincere gratitude to the reviewers for their insightful feedback, which has significantly enhanced the quality and clarity of this manuscript. We hope the revised version satisfactorily addresses the concerns raised.

The reviewers' primary requests focused on two key areas:

1. Providing additional controls for RNAi knockdowns of target genes.
2. Tempering our claims regarding the specific amyloid assemblies (oligomers, intermediates, fibrils, etc.) responsible for toxicity in the *C. elegans* and mice models.

Our point-by-point responses are detailed below:

1. RNAi Knockdown Efficiency and Controls

We have included new data confirming the efficiency of the various RNAi strategies employed in our worm models (**SF1A**, **SF1E**). These results consistently demonstrate a significant reduction in target RNA levels. Furthermore, Western blots documenting the silencing efficiency of the RNA Exosome 10 in mouse brains are provided in **SF3A** and **SF3E**. These data illustrate effective silencing, with the mouse-to-mouse variability observed being consistent with standard results for *in vivo* mammalian experiments.

2. Clarification of Amyloid Assemblies and Toxicity

As noted in our original cover letter, the core conceptual advance of this work is the link between enzymes of physiological amyloidogenesis and pathological amyloidogenesis-bridging two research fields that have historically evolved in isolation. Our objective was not to take a definitive stance on the ongoing debate regarding specific toxic species.

To ensure our conclusions remain appropriately tempered and focused on our primary findings, we have implemented the following changes:

- A) Removed the models previously presented in **SF1A** and **SF3**.
- B) Deleted all specific mentions of "oligomers," "intermediates," and "fibrils" from the Abstract and Main Text.
- C) Revised the Discussion to explicitly reflect a neutral position regarding the specific nature of the toxic assembly.

We believe these revisions maintain the integrity of our findings while respecting the complexity of the current discourse in the field.

Reviewer 1

We thank the reviewer for their encouraging and constructive comments. We have addressed each point and revised the manuscript accordingly. Notably, we have adjusted our conclusions regarding the toxic forms of amyloids to align with the reviewer's request (also other reviewers and Editor) for a more tempered interpretation. We have kept original datasets regarding toxic forms of A β and aggregation of α -synuclein in worms while removing mentions of oligomers and fibrils.

Comment #1. In the case of α -synuclein in worms, some evidence of aggregation is provided in the form of confocal microscopy and analysis of aggregate number and sizes. However, no evidence is provided that A β behaves similarly. Indeed, when amyloid plaque size is assessed in mice in Fig 3, there is no difference in plaque size. Figure 1C shows a partial rescue of *gld-4* and *gld-2* by O4 which drives fibrillation and suppresses toxicity, however this is an orthogonal experiment which demonstrates that reducing toxic species via O4 driven fibrillation mitigates toxicity but doesn't otherwise demonstrate the mechanism of action discussed. Other interpretations such as removal of toxic oligomers would also explain the observations provided for TENTs and RNA exosome. Furthermore, the α -syn puncta may not represent amyloids (in our experience these are often soluble), or even insoluble proteins and this needs to be further explored. Therefore, while the authors propose that formation of non-toxic amyloids underlies the reduction in toxicity, there has not been a sufficient investigation of the formation of these amyloids in the context A β and α -syn.

Response. To the best of our knowledge, current literature suggests that the toxic form of A β^{1-42} in the *C. elegans* model CL2006 corresponds to the approximately 15 kDa bands observed on Western blots. Our data (**F1B,F2B**) (see below for an example) indicate a correlation between the appearance of these bands in *gld-4* and *gld-2* loss worms and their reduction in *crn-3* loss worms, which may account for the observed variations in toxicity.

While the exact nature of the 15 kDa band remains to be fully elucidated as oligomers or fibrils—its presence correlates with toxicity in the CL2006 model. We have modified the text to temper our conclusions regarding the biochemical properties of this 15 kDa species, specifying that it is widely regarded in the literature as the toxic form of A β^{1-42} in CL2006. Similarly, we have revised the manuscript to remove definitive claims regarding the toxicity of α -synuclein aggregation levels in the NL5901 model.

Example of Deleted Text to remove words ascribing toxicity to oligomers, intermediates, fibrils, etc.): Abstract, Line 33:

In contrast, the RNA exonuclease Exosc10 potentiates pathological amyloid toxicity ~~associated with the accumulation of amyloidogenic intermediates.~~

Example of New Text to temper down claims of toxic amyloid assemblies. Line 177:

While we cannot formally conclude that TENT4b/2 activity or Exosc10 loss reduce β -amyloid¹⁻⁴² toxicity through conformational changes in *C. elegans* and mice models, we note that this correlation is consistent with reports of oligomeric toxicity of β -amyloid¹⁻⁴² [32, 38, 43].

Example of data showing the multimeric β -amyloid¹⁻⁴² form believed to be toxic in the CL2006 model (arrow) (F2B).

Comment #2. It is observed that inhibition of *gld-4* and *gld-2* both increase paralysis. It appears that *gld-4* inhibition decreases "aggregation" of α -syn, in line with the paper's conclusion that TENTs facilitate the formation of non-toxic amyloid species. However, *gld-2* RNAi displays the opposite action on α -syn aggregation, suggesting that its wild type action opposes the formation of aggregates, appearing to contradict the conclusions that TENTs in general facilitate the formation of non-toxic amyloid aggregates, both because GLD-2 does not appear to do this and because in the *gld-2* RNAi context α -syn aggregates appear to be associated with higher toxicity.

gld-2 should be moved to the supplement or more thoroughly investigated.

Clearer discussion of *gld-2* in general in relation to the different mechanism.

Response. Following the reviewer's recommendation, we have moved the *gld-2* experiments to the Supplemental Information. We have also added text to address the discrepancy in aggregation patterns, noting that unstable α -synuclein aggregates can be associated with toxicity—a phenomenon that remains challenging to definitively test in *C. elegans* models.

New Text, Line 180:

TENT2^{GLD-2} loss-associated α -synuclein aggregation suggest the involvement of alternative toxicity mechanisms, including the formation of aggregate more prone to shedding [43, 44], a phenomenon that remains challenging to definitively test in *C. elegans*.

Comment #3. Intro - Previous studies have shown that TENT4b facilitates formation of mature amyloidogenic aggregates, however this seems to only have been shown in nuclear localised A-bodies, and A β and α -syn mostly aggregate elsewhere. Some discussion of the potential mechanism by which RNA tailing of long-noncoding RNA derived from the ribosomal intergenic spacers would influence disease protein aggregation in the cytoplasm should exist in the text.

Response. We appreciate this insightful observation. Our model proposes that ribosomal intergenic spacer lncRNA (rIGSRNA) recruits TENT4b to the nucleoli to drive aggregation. We have also observed that the depletion of RNA Exosome 10 (*crn-3*) triggers aggregation not only within nucleoli but also in other cellular compartments, a process partially dependent on TENT4b. This suggests that TENT4b, and potentially other TENTs, may facilitate aggregation across different subcellular domains opposed by RNA Exosome. Furthermore, it is established that A β and other pathological amyloids can aggregate within nucleoli both in vitro and in vivo. The revised text now includes a discussion of these points.

New Text, Line 68:

...RNA Exosome antagonizes this process by inhibiting intracellular amyloidogenesis, both within the nucleolus as well as in the cytoplasm.

New Text, Line 81:

Depletion of the RNA Exosome catalytic subunit EXOSC10 removes this inhibition, enabling the formation of mature amyloidogenic aggregates in different subcellular compartments.

Comment #4. Line 112 - *gld-2* & *gld-4* inhibition is said not to change A β transgene expression, but *gld-2* significantly decreases it. This should be mentioned in the text.

Response. We have updated the text to explicitly discuss the decrease in A β transgene expression observed with *gld-2* inhibition.

New Text, Line 106:

RNAi depletion of *TENT2*^{*gld-2*} also resulted in increased paralysis (**Fig.S1A,C**) and the accumulation of toxic β -amyloid¹⁻⁴² species in CL2006 *C. elegans* (**Fig.2B**) even if it decreased β -amyloid¹⁻⁴² transgene levels (**Fig.S1B**).

Comment #5. Figure 2 D,E,F,G: Should show gld-2, gld-4 controls as well as doubles with crn-3.

Response. These data are provided in **SF1C** to avoid overcrowding the main figures.

Comment #6. Fig S1C - A crn-3 alone control would be helpful here.

Response. The crn-3 alone control is displayed in **F2A** for direct comparison with the experimental groups.

Comment #7. Figure S1B legend - show the n= for the qPCR.

Response. We have included the n-values for all qPCR experiments, including the newly added data, in the figure legends.

Comment #8. Figures general - Clearer labelling of gels which show monomer/soluble/oligomeric A β to show what the bands are said to be.

Response. We have added additional arrows to the relevant gel bands. However, in accordance with the reviewers' request to remain conservative in our terminology, we have refrained from using specific terms like "oligomers".

New Text, Line 384 and 398:

Arrow indicates soluble multimeric species believed to be the toxic form of β -Amyloid

Comment #9. Fig 3 panels should be rearranged to match with the order they appear in the text.

Response. Figure 3 and the corresponding text have been reordered for consistency.

Comment #10. Line 463 references Amylo-Glo staining, but this is not shown in the paper.

Response. We have corrected the text to remove the erroneous reference.

Comment #11. More detail on the methods for brain & worm lysates should be shown in methods section. More detail on intranasal administration of the treatment in mice, including buffers used & how long between end of regime and behavioral testing.

Response. We have expanded the Materials and Methods section to include comprehensive details regarding the preparation of brain and worm lysates, the buffers used for intranasal administration in mice, and the specific timeline between treatment and behavioral assessments.

New Text, Line 514:

For mice, hippocampus and cerebral cortex were rapidly dissected in ice-cold phosphate-buffered saline, snap-frozen, and stored at -80° C before use. Brain regions were homogenized in ice-cold Triton-lysis buffer 20 mM Tris-HCl, pH 7.4, 150 mM NaCl, 1 mM EDTA, 1 mM EGTA, 1% Triton-100, 10 μ g/ml leupeptin, 10 μ g/ml aprotinin, 5 μ g/ml pepstatin, 1 mM phenylmethanesulfonyl fluoride, 1 mM sodium vanadate, 50 mM sodium fluoride, and 100 nM okadaic acid. The lysates were centrifuged at 15,000 g for 10 min to remove insoluble debris, and protein concentrations in the supernatants were determined using the Bradford protein assay. Proteins (10–20 μ g) in brain region extracts were resolved with SDS-PAGE, transferred to nitrocellulose membranes, and immunoblotted with primary antibodies.

For CL2006, worms were synchronized and plated onto appropriate RNAi plates for 2 generations. ~200-500 Day 3 old adults were collected, lysed in (50mM Tris/HCl pH 8.0, 0.5M NaCl, 4mM EDTA, 1% NP40) using sonication (50% power for 10 sec, 6 times) spun at 14000G for 10 min. Equal concentration of protein was loaded for western blots and was probed with APP-6E10 (Novus, NBP2-62566). Age is defined as days past L4 stage. For all primary antibodies used (1:1,000) were the following: APP-6E10 (Novus, NBP2-62566), Exosc10 (Santa

Cruz, sc-374595), beta-Actin (Life Tech, MA5-15739). For 6e10 blots, PVDF membrane was submerged in boiling PBS for 5 min prior to incubating with blocking buffer.

New Text, Line:541:

Mice were injected intranasally with 10 ug of scramble siRNA (#5, AM4642 Ambion), or ExoSC10 siRNA (J-049286-05, Dharmacon) in PBS, and 5uL were administered in each nostril every other day for a week consistent with [85–87] and the behavior was initiated on the 3rd day (open field), and on the 6th day (Y maze and novel object recognition).

Reviewer #2

We thank the reviewer for their constructive and encouraging comments. We have tempered our claims regarding the toxic forms of amyloids within our systems and hope that the reviewer will find the revised manuscript satisfactory.

Comment #1. This paper largely confirms a consensus opinion in the amyloid field that the main toxic species in protein aggregation diseases are soluble oligomers, while insoluble fibrils are less toxic. Their data are generally consistent with and confirm this opinion. They tend to overstate or state too starkly the difference in toxicity of oligomers and fibrils, however. It's not quite as all-or-nothing as they suggest. In some studies, fibrils do have some toxicity.

Response. We agree with the reviewer that certain claims were overstated. Accordingly, we have removed the model previously shown in **SF1A** and revised the text in both the Abstract and the main manuscript. These changes adopt a more neutral position, acknowledging that our datasets do not allow for a definitive conclusion regarding which specific amyloid species are toxic.

Example of Deleted Text to remove words ascribing toxicity to oligomers, intermediates, fibrils, etc.): Abstract, Line 33:

In contrast, the RNA exonuclease Exosc10 potentiates pathological amyloid toxicity ~~associated with the accumulation of amyloidogenic intermediates.~~

Example of New Text to temper down claims of toxic amyloid assemblies. Line 177:

While we cannot formally conclude that TENT4b/2 activity or Exosc10 loss reduce β -amyloid¹⁻⁴² toxicity through conformational changes in *C. elegans* and mice models, we note that this correlation is consistent with reports of oligomeric toxicity of β -amyloid¹⁻⁴² [32, 38, 43].

Comment #2. Similarly, their work tends to confirm a growing consensus linking RNA metabolism to amyloid formation. Thus, TENT4b and TENT2 tend to abrogate toxicity, while exonucleases increase it. However, while their data are generally valid and support their claims, some of the details of their studies are more ambiguous than they acknowledge. In Figure 1, for example, the difference between EV and *gld4* (panel E and even more so panel F) is statistically significant but there is a lot of overlap in the two groups. In panel H, the difference between *gld2* and either EV or *gld4* seems to depend entirely on two outlier points in *gld2* worms. Similar comments apply to some of the data shown in Figure 2. A more nuanced approach to such data is warranted. While the statistical significance exists, they could be more forthcoming about limitations in their data.

Response. We have incorporated new text to explicitly address the limitations of our datasets. Specifically, we have added clarifying remarks regarding the data distribution and variability as follows:

New Text, Line 101.

The protective effect of TENT4/Gld-4 was not limited to β -amyloid¹⁻⁴², as depletion of *TENT4^{gld-4}* moderately suppressed aggregation and increased toxicity of α -synuclein-YFP in *C. elegans* model of Parkinson's disease,

NL5901[34] (**Fig.1D-H**). While increased variability of the data exists, these results indicate that *TENT4^{GLD-4}* protects against amyloid-induced toxicity from two distinct amyloidogenic peptides.

Comment #3. What kind of gel is shown in Figure 2B (and elsewhere)? This should be commented upon.

Response. We have added descriptive text to the Figure Legends and the Materials and Methods section as follows:

New Text, Line 381:

(B) *TENT4^{GLD-4}* suppresses soluble β -Amyloid. SDS-PAGE western blots showing soluble β -amyloid (6E10 antibody) in CL2006 following RNAi against *TENT4^{GLD-4}*

New Text, Line 397:

(B) SDS-PAGE western blots showing soluble β -amyloid (6E10 antibody) in CL2006 following RNAi against *TENT4^{GLD-4}*, *TENT2^{GLD-2}* and *ExoSC10^{CRN-3}*.

New Text, Line 513:

Immunoblot. For mice, hippocampus and cerebral cortex were rapidly dissected in ice-cold phosphate-buffered saline, snap-frozen, and stored at -80° C before use. Brain regions were homogenized in ice-cold Triton-lysis buffer 20 mM Tris-HCl, pH 7.4, 150 mM NaCl, 1 mM EDTA, 1 mM EGTA, 1% Triton-100, 10 μ g/ml leupeptin, 10 μ g/ml aprotinin, 5 μ g/ml pepstatin, 1 mM phenylmethanesulfonyl fluoride, 1 mM sodium vanadate, 50 mM sodium fluoride, and 100 nM okadaic acid. The lysates were centrifuged at 15,000 g for 10 min to remove insoluble debris, and protein concentrations in the supernatants were determined using the Bradford protein assay. Proteins (10–20 μ g) in brain region extracts were resolved with SDS-PAGE, transferred to nitrocellulose membranes, and immunoblotted with primary antibodies. For CL2006, worms were synchronized and plated onto appropriate RNAi plates for 2 generations. ~200-500 Day 3 old adults were collected, lysed in (50mM Tris/HCl pH 8.0, 0.5M NaCl, 4mM EDTA, 1% NP40) using sonication (50% power for 10 sec, 6 times) spun at 14000G for 10 min. Equal concentration of protein was loaded for western blots and was probed with APP-6E10 (Novus, NBP2-62566). Age is defined as days past L4 stage. For all primary antibodies used (1:1,000) were the following: APP-6E10 (Novus, NBP2-62566), Exosc10 (Santa Cruz, sc-374595), beta-Actin (Life Tech, MA5-15739). For 6e10 blots, PVDF membrane was submerged in boiling PBS for 5 min prior to incubating with blocking buffer.

Comment #4. There are many small errors in the language. They should do a thorough proofreading before resubmitting their manuscript.

Response. We have performed a comprehensive proofreading of the manuscript and corrected the linguistic errors identified.

Comment #5. The main limitation of this paper is that while it expands some of the phenomenology of amyloid-related diseases, it really does not get into mechanism at all. Exactly how might RNA alter the course of these diseases. Do they mean any and all RNA? Why enzymes affecting RNA tailing in particular? Is the effect of RNA direct (e.g., by binding β -amyloid or α -synuclein) or indirect? If the former, how generalizable is the finding? If the latter, what other players are involved.

Response. The primary objective of this manuscript is to bridge the pathways of physiological amyloidogenesis with those of pathological amyloids—two research fields that have largely evolved independently. We believe the current study successfully establishes this link. While the TENT/RNA exosome machinery regulates a diverse array of RNAs and pathways, elucidating the precise molecular mechanisms by which these pathways modulate amyloid toxicity remains a significant challenge. We feel that such a mechanistic investigation lies beyond the scope of the present study but represents a vital direction for future research.

Reviewer #3

We thank the reviewer for their constructive comments. We have addressed each point individually and hope that the reviewer will be satisfied with the revised version of the manuscript.

Comment #1. The specificity of RNAi here is not demonstrated for any of the genes targeted either in *C. elegans* or mice. For example, does *crn-3* RNAi treatment impact *crn-3* transcript levels and/or protein abundance.

Response. To address this, we have performed RT-qPCR on RNA isolated from worms treated with RNAi targeting *gld-4*, *gld-2*, and *crn-3*. These new data have been incorporated into **SF1A,E**.

New Data: SF1A,E.

Comment #2. This study is heavily dependent upon RNAi methodology and body wall muscle expression of amyloid which has been shown to not necessarily translate well to neurodegeneration in AD.

Response. We agree with the reviewer regarding the inherent limitations of using body wall muscle models to study Alzheimer's Disease (AD) neurodegeneration. Indeed, murine models also possess limitations in recapitulating human disease. The primary objective of this manuscript is to bridge the fields of physiological amyloidogenesis and pathological amyloids—two areas of research that have evolved largely independently over the years. We believe the current work achieves this goal. Determining whether the TENT/RNA Exosome machinery is directly involved in human neurodegenerative diseases remains a compelling challenge for future investigations and lies beyond the scope of this study.

Comment #3. While the number of animals studied for histology seems marginally sufficient, the behavioral studies seem underpowered, especially given the lack of validation and measurement of siRNA treatment in rodents.

Response. Western blot analysis of siRNA silencing efficiency for the RNA Exosome 10 experiments in the murine brain is provided below and have been included in **SF3A,E**. The data demonstrate efficient silencing, with the mouse-to-mouse variability typically expected in *in vivo* experiments. It should be noted that these β -amyloid toxicity models—particularly the aged 3xTg model—are resource-intensive to acquire and maintain. Consequently, we utilized a cohort size sufficient to achieve statistical significance while adhering to our institutional budgetary constraints.

Western showing decrease in RNA Exosome protein levels by westerns in 5XFAD mice treated with siRNA against ExoSC10 (SF3A).

Comment #4. Including validation of at least the main findings using a neuronal amyloid model would be more disease relevant and increase the impact of this work. An ideal experiment would be to recapitulate the RNAi findings from muscle instead in neurons using genetic ablation of the genes in question. This could be accomplished using existing *crn-3* conditional or partial loss of function alleles.

Response. We appreciate this insightful suggestion. As we did not sufficiently explain the rationale of using the CL2006 strain. We utilized the CL2006 β -amyloid toxicity model as it remains the historical standard in the field. Crucially, the toxicity timeline in this model allows us to accurately measure both the accelerated toxicity resulting from *gld-2/4* RNAi and the delayed toxicity following *crn-3* RNAi. Other *C. elegans* models, including various neuronal models, exhibit rapid toxicity that precludes a clear assessment of *gld-2/4* RNAi effects. Furthermore, the β -amyloid mouse models provide a more robust and relevant system for studying neuronal effects than nematode models.

We have added text in the manuscript to make this point clearer.

New Text, Line 86:

To address this question, we first chose the classical *C. elegans* strain (CL2006) that constitutively expresses Alzheimer's disease (AD)-relevant human β -amyloid¹⁻⁴² [27]. The CL2006 transgenic *C. elegans* undergo age-dependent paralysis owing to the accumulation of β -amyloid¹⁻⁴² in body wall muscle cells [28] and have been used extensively to uncover biochemical pathways and small molecules involved in β -amyloid¹⁻⁴² intracellular toxicity [29]. This model was chosen because the toxicity timeline allows to accurately measure both the accelerated and decreased toxicity resulting in manipulated the TENT/RNA Exosome system. In addition, other systems which induce β -amyloid¹⁻⁴² expression by heat induced stabilization of the transgene introduces confounding effects of the RNA tailing and degradation machinery on β -amyloid¹⁻⁴² mRNA

February 20, 2026

RE: Life Science Alliance Manuscript #LSA-2025-03493-TR

Prof. Stephen Lee
University of Miami
Biochemistry and Molecular Biology
1011 NM 15th Street, Room 217
Miami, Florida 33136

Dear Dr. Lee,

Thank you for submitting your revised manuscript entitled "Enzymes of Physiological Amyloidogenesis Control Pathological Amyloid Toxicity". We returned this to Reviewer 1 who is satisfied with no further requests, and we feel the remaining reviewer points have been similarly fully addressed. We would be happy to publish your paper in Life Science Alliance pending final revisions necessary to meet our formatting guidelines.

MANUSCRIPT ORGANIZATION AND FORMATTING:

To avoid unnecessary delays in the acceptance and publication of your paper, please read the following information carefully. Full guidelines are available on our Instructions for Authors page, <https://www.life-science-alliance.org/authors>

- Please upload your main manuscript text as an editable doc file.
- Please upload all figure files as individual ones, including the supplementary figure files; all figure legends should only appear in the main manuscript file.
- Please add ORCID ID for corresponding author - you should have received instructions on how to do so.
- Please add a Summary Blurb/Alternate Abstract in our system.
- Please add the X and Bluesky handles of your host institute/organization, as well as your own, and/or one of the authors, in our system.
- We encourage you to revise the figure legends for Figure S1 such that the figure panels are introduced in alphabetical order.
- Please consult our manuscript preparation guidelines <https://www.life-science-alliance.org/manuscript-prep> and make sure your manuscript sections are in the correct order.
- Please describe the objectives used for fluorescence imaging in the Methods section (manufacturer, type, magnification, NA).
- Please add an Author Contributions section to your main manuscript text.
- Please add a Conflict of Interest statement to your main manuscript text.
- Please indicate the scale bar size in the legend for Figure 3, related to panel F, and please also add a scale bar to Figure S3G. Please indicate the channels shown in the legends for each.

LSA encourages authors to provide a 30-60 second video where the study is briefly explained. We will use these videos on social media to promote the published paper and the presenting author (for examples, see <https://docs.google.com/document/d/1-UWCfbE4pGcDdcgzcmiuJl2XMBJnxKYeqRvLLrLS08s/edit?usp=sharing>). Corresponding or first-authors are welcome to submit the video. Please submit only one video per manuscript. The video can be emailed to contact@life-science-alliance.org

FINAL FILES:

To upload the final version of your manuscript, please log in to your account: <https://lsa.msubmit.net/cgi-bin/main.plex>

The following items are required for acceptance.

The license to publish form must be signed before your manuscript can be sent to production. A link to the license to publish form will be available to the corresponding author only. Please take a moment to check your funder requirements.

Thank you for your attention to these final processing requirements. Please revise and format the manuscript and upload materials as soon as you are able.

Thank you for this interesting contribution to the literature. We look forward to publishing your paper in Life Science Alliance.

Sincerely,

Reviewer #1 (Comments to the Authors (Required)):

This is a very interesting paper with an important research question - the authors investigate whether RNA tailing dynamics, which they have previously shown modulate the maturation of aggregates, is important for modulating toxicity of human disease related proteins A β and α -syn. This paper very effectively shows that wild type function of TENT4, and inhibition of the opposed RNA exosome each reduce toxicity of these disease-associated proteins. In particular, the mouse data in figure 3 is very striking and shows a strong removal of oligomeric A β in a mouse model, along with associated improvements in memory.

I believe the concerns I raised have been adequately addressed by the authors, and that the article is now suitable for publication.

March 3, 2026

RE: Life Science Alliance Manuscript #LSA-2025-03493-TRR

Prof. Stephen Lee
University of Miami
Biochemistry and Molecular Biology
1425 NW 10th Ave, Miami, FL 33136. Room 1005
Miami, Florida 33136

Dear Dr. Lee,

Thank you for submitting your Research Article entitled "Enzymes of Physiological Amyloidogenesis Control Pathological Amyloid Toxicity". It is a pleasure to let you know that your manuscript is now accepted for publication in Life Science Alliance. Congratulations on this interesting work.

Your article will publish open access upon publication under a CC-BY license.

DISTRIBUTION OF MATERIALS:

Again, congratulations on a very nice paper. I hope you found the review process to be constructive and are pleased with how the manuscript was handled editorially. We look forward to future exciting submissions from your lab.

Sincerely,
